# Entrepreneurship and Sport: A Strategy for Social Inclusion and Change

**DOI:** 10.3390/ijerph18094720

**Published:** 2021-04-28

**Authors:** Giuseppina Maria Cardella, Brizeida Raquel Hernández-Sánchez, José Carlos Sánchez-García

**Affiliations:** Department of Social Psychology and Anthropology, University of Salamanca, 37005 Salamanca, Spain; brizeida@usal.es (B.R.H.-S.); jsanchez@usal.es (J.C.S.-G.)

**Keywords:** scientific mapping analysis, sport for development, social inclusion, social change, sport entrepreneurship, bibliometric analysis

## Abstract

The relationship between sport and entrepreneurship is an emerging but constantly evolving research field, especially in recent years. This is an extremely important topic considering its positive impact on public health, social inclusion, economic development, and the promotion of intercultural exchange. This study has carried out a comprehensive and systematic review of literature on entrepreneurship and sport as tools for implementing social change. We used several bibliometric indicators in order to understand the current state of the literature, and scientific mapping analysis (keywords co-occurrence) to interpret the main theoretical areas of scientific interest. By searching the Scopus database, a total of 239 peer-reviewed articles were analyzed. The results showed the identification of six major recurring themes in the literature. For the purposes of our contribution, the recently developed line of research which considers sport as a tool for solving social problems through social change appears to be of particular importance. These data allow us to conclude that it is a highly multidisciplinary and active field. Suggestions for future lines of research and methodological limitations are discussed.

## 1. Introduction

Entrepreneurship is a process aimed at social transformation [1,2], economic growth [3], the supply of services and new products through the stimulation of creativity, and the production of innovative ideas [4].

Entrepreneurship is a career of interest given how it helps to overcome economic crises, potentiating the development of a strong economy and new job opportunities. This is a growing area of interest among academics and researchers from different disciplines, becoming a source of discovery and the exploration of opportunities [5].

The issue of interest for this study, sports entrepreneurship, is understood as an emerging and constantly evolving area of the business and management world, involving innovative social activities that are carried out in a sporting context for the creation of new markets [6] and social inclusion [7]. As stated by Stier [8], possession of the entrepreneurial knowledge and skills necessary to face the challenges, especially in the sports industry, seems to be a mandatory requirement for those who manage sports organizations.

Because of this, analyzing the sports environment through entrepreneurial processes is useful in improving the well-being and performance of sports companies [9,10], as well in increasing their economic production [11].

According to Peredo and Chrisman [12], an entrepreneurial approach in sport could be used as a catalyst to mitigate the current situation of economic decline, and identify new opportunities that are of considerable importance in the value creation process.

In addition, Legg and Gough [13] stressed the importance of an entrepreneurial strategy, especially in hostile sports contexts, to transform sports organizations into professional and highly competitive companies.

In this sense, according to Vamplew [14], sports entrepreneurs are considered to be agents of change who operate in the sports sector, trying to increase productivity, improve their experience, and increase interest in products and services, generating the development of new markets.

Historically, the first theoretical contributions to sport entrepreneurship date back to Hardy [15], who proposed a tripartite model for the analysis of entrepreneurship in sport, as sport is an industry with special characteristics. Specifically, according to the author, the essential components of sports entrepreneurship are: the sports product, the role that entrepreneurs and managers play in the product development phase, and the types of organizations created.

In addition, sports entrepreneurs adapt well to the model proposed by Schumpeter [16], since the skills commonly associated with the classic entrepreneur—such as risk taking, proactivity, innovation, and the pursuit of profits—also find a broad consensus in the field of sport [17]. An example can be seen in professional soccer teams, which can be considered true small and medium enterprises due to some of their organizational and management characteristics, such as annual turnover and the number of employees [18].

In recent years, Hardy’s tripartite model has been adopted by Porter [19], who used it as a basis for exploring different types of sporting activities, introducing a new element—namely, the importance assumed by the media in the dissemination of sport—and placing special emphasis on the entrepreneurial process as a tool for the creation of networks and connections, exploiting new combinations of goods and services. For Porter [19], this tripartite model must be the starting point of any study worth mentioning in entrepreneurship in sport.

Despite the unquestionable importance of entrepreneurship in the field of sport, there are only a few entrepreneurs who, despite having sports knowledge and skills, seek job opportunities in this field [20,21]. This lack of entrepreneurial activity in the sports sector has shifted the focus of academics towards a line of research aimed at supporting the development of entrepreneurial skills in the younger generations, especially in university students [22,23,24,25] and socially excluded youth [26].

For example, Sport for Development (SFD) programs can be tools that support inclusive action. Socially disadvantaged groups can be reached more easily, because socio-cultural barriers are removed, and because participants are actively recruited [27]. Furthermore, as empirical evidence shows, Sport for Development has a positive influence on personal development, resilience, self-esteem, and social skills [28,29,30]. These important achievements can further increase job opportunities and other forms of social inclusion, especially among young people.

Taking into account these different points of reflection on entrepreneurship and sport, the objective of this work is to analyze the empirical research and theoretical development on the subject, in order to obtain an impression of the current state of research, its main lines, and their strengths and weaknesses, as well as to identify useful topics in order to project future lines of work in the field.

To achieve these objectives, we set out to answer the following questions: (1) How has the relevant scientific research developed over the years? (2) Which are the most productive authors, countries, and journals in the scientific landscape? (3) What are the most important issues identified by the existing literature on entrepreneurship in a sports context? (4) Which methodological approaches have been used the most? (5) What are the possible gaps currently present in the scientific literature on the relationship between entrepreneurship and sport, and what are the possible lines of interest for future research?

In the following sections we explain the methodology used and the main results. In the final section, we present the conclusions that can be drawn from our analysis, their limitations, and grounds for future developments.

## 2. Materials and Methods

We aim to contribute to the systematization of scientific production on the relationship between entrepreneurship and sport. The SCImago Group’s Scopus database was used to search for scientific articles. Based on previous research and the results obtained, we decided to choose Scopus, since it is the largest database of scientific literature (27 million abstracts), and a curated, high-quality source of bibliometric data in the form of quantitative scientific studies [31,32].

The selected search terms have included the words “entrepren*” and “sport”, using the “AND” Boolean connector, and including as a search field “all fields”, without time margins.

The bibliographic search ended in February 2021, generating a total of 599 articles.

After excluding articles written in languages other than English or Spanish, presentations at conferences, book chapters, theses, and all articles related to the year 2021, we proceeded to read the abstract of each article. When an abstract generated doubts or conflicting opinions, we read the full article.

The final selection of the articles was made by applying the following criteria: (1) scientific articles published in peer-reviewed journals (including press articles), as they are considered valid sources of knowledge [33]; (2) written in English or Spanish; (3) published up to the year 2020; and (4) demonstrating the importance of, and direct relationship between, entrepreneurial processes and sports (for example, all articles related to the topic of tourism were excluded).

This selection phase produced a final result of 239 articles.

To minimize the subjective component and possible attribution errors, the guidelines of the PRISMA (Preferred Reporting Items for Systematic Reviews and Meta-Analyses) method [34,35,36] were adopted, and a series of bibliometric indicators were used to analyze the temporal evolution of scientific production, the most influential authors, the most productive scientific journals with reference to the number of articles published, the countries with the greatest number of scientific contributions, the methods of analysis adopted (quantitative, qualitative research, and non-empirical), and the sample(s) used. In this section, in order to measure the impact of the author and the journal, the citation of Scopus and the h-index, developed by the physicist Hirsch [37], were used. As the author points out, a scientist has an *h*-index if he has published *n* articles with at least *n* citations each.

For science mapping analysis, VOSviewer software, version 1.6.10 [38,39], was used to analyze the most prevalent and emerging topics within the sport entrepreneurship knowledge base. This is an indisputable statistical technique [38], currently in widespread use [40,41,42,43]. Based on similarity and co-occurrence techniques (keywords used by the authors), this method allows us, through the construction of maps, to investigate the type and strength of the relationships between the different areas of scientific interest. In addition, each area is associated with a different cluster marked with a specific color.

Figure 1 shows the flow chart of the bibliographic research according to the recommendations of the PRISMA method.

## 3. Results

### 3.1. Bibliometric Analysis

Figure 2 shows the progress of scientific research on entrepreneurship and sport. As can be seen, this is a fairly recent research field (the first article dates from 2004), which registered a significant increase in recent years, reaching its highest publication peak in 2020.

This increase, specifically in recent years, could suggest a change of interest in scientific research, and a continuous and growing evolution of entrepreneurship in sport as a trending issue.

Cluster analysis (keywords map) was carried out in order to analyze the thematic areas of greatest interest in the scientific landscape. Using VOSviewer software [38,39], with a minimum of three co-occurrences per keyword, it was possible to associate the “entrepreneurship” and “sport” constructs with six different lines of research, which underlines the multidisciplinary nature of the different lines of research in relation to the 2004–2020 time period.

Specifically, although the topic of entrepreneurship and sports research has been of broad scientific interest, covering the entire time frame (Cluster 4), other areas have achieved their greatest development only during certain periods, registering a sudden decrease in recent years.

For example, Cluster 3, related to the area of social entrepreneurship, had its highest peak in 2017, while Cluster 5 reached its highest number of publications in 2015, but in recent years both have seen a significant reduction of interest from the scientific community.

The area of study that received the least consensus is related to Cluster 6, associated with the keywords “change” and “stakeholders”, with a low trend over time, followed by education and sports marketing (cluster 2), which had a greater scientific expansion, especially during the first decade of 2000.

Cluster 1, related to sport entrepreneurship, deserves a separate discussion. Interest in this area seems to have grown in recent years (2017), and it seems to be in potential evolution. In fact, unlike the other clusters, it is the only one in which interest is currently increasing (Figure 3).

From this analysis we can deduce that, historically (early 2000s), sport entrepreneurship has been studied only as a branch of other constructs that enjoyed greater importance, such as those of social entrepreneurship and entrepreneurial intention; however, in recent years, sport entrepreneurship as an independent research topic is receiving increasing attention from the scientific community.

Focusing on the productivity of scientific journals, we have taken into account those that generated a minimum of five publications in the field. In Table 1 we present the main results, classifying the scientific articles from the most productive to the least productive. The 12 most productive journals published 98 articles, representing 42.6% of the full sport entrepreneurship knowledge base. Our analysis also allowed us to analyze the most associated research areas (Table 1).

More than half of the articles were published in journals related to business and management (administration) (39%) and social sciences (36.6%); however, we also found other areas in journals related to sport science (29.1%) and economic science (6.5%). This suggests that the sport entrepreneurship field is multidisciplinary, and encompasses many types of organizations in many sectors. In addition, it highlights the importance that entrepreneurship plays in the sports field as a tool for the generation of competitive advantages in the global market.

We also carried out an analysis of the authors, identifying those who published the most studies in the field of entrepreneurship and sports. In the 239 articles selected for the bibliometric analysis, a total of 449 authors were found, with an average of less than 2 authors per article (1.87). In addition, only 12% of the authors contributed to more than two published articles, which shows that this is a fragmented field of research with poorly defined limits and characterized by poor collaboration among researchers, probably due to its relatively recent development in the scientific landscape (Table 2).

The data trend in Figure 4 shows that sport entrepreneurship publications are centered in the USA, UK, Australia, Spain, and Canada. Scholars located in these five Western countries produced 173 of the 239 articles (58%) from the Scopus-indexed knowledge base on sport entrepreneurship.

This knowledge production trend is in line with the trend in the management literature, in which Anglo–American scholars have dominated international publications in the English language [44].

Furthermore, a deeper analysis shows low involvement of developing countries (9%) in the sports entrepreneurship scholarship. This is a shortfall that should not be overlooked, especially given the potential positive impact (in the social and economic sectors) that sports entrepreneurship activities have in these disadvantaged countries, where problems are more likely to be solved by initiatives promoted by citizens [45].

Lastly, we performed an analysis of the type of research carried out and the samples most frequently used. The frequency analysis showed a low number of publications in the fields of quantitative research (32.6%) and non-empirical scientific contributions (28.3%). A qualitative survey method was used in 44.6% of the articles analyzed, mainly through case studies and in-depth interviews. Only 2.4% of the articles (*n* = 3) used a mixed approach (both quantitative and qualitative).

Regarding the type of sample, we have grouped the different categories into three groups: (1) sports entrepreneurs/sports managers, who represent the group most frequently used in the three types of analysis; (2) university sports students, who were most frequently used in quantitative research; and (3) athletes, with the highest frequency of use in qualitative research (Table 3).

### 3.2. Topic Research on Sport Entrepreneurship

To get an overview of the main lines of research in the field of entrepreneurship and sport, a cluster analysis was used, using one co-occurrence for each keyword, for a total frequency of 287 words grouped into 38 clusters.

The research topics most analyzed by academics relate to entrepreneurship, sport, but also to sports entrepreneurship, social entrepreneurship, innovation, education, management, and entrepreneurial intentions. As shown in Figure 5, these keywords are represented by larger circles, indicating greater strength and intensity in the relationships between different groups.

Given the multidisciplinary nature of entrepreneurship and sport, and in order to reduce the field of analysis, an analysis with a minimum of 3 co-occurrences per keyword was conducted, for a total of 22 keywords grouped into 6 clusters (Figure 6).

As previously mentioned, mapping and grouping provide us with information on the importance attributed by academics to the field of sport entrepreneurship. In addition, each group has a different color that emphasizes the different relationships between the various areas, while the distance between the groups shows the intensity of the relationships [36]. Below, we show a summary table of the clusters and the associated keywords (Table 4).

#### 3.2.1. Cluster 1: Sport Entrepreneurship

The first cluster, red, is made up of the following keywords: sports entrepreneurship, entrepreneurial orientation, social capital, sports management, and innovation. 19.8% of the keyword co-occurrences are part of this group.

This area represents the heart of our study, as it highlights the intensity of the cluster’s relationship with other clusters, such as entrepreneurship, sports, and social entrepreneurship, which appear to be the most influential areas in our analysis. However, it occupies a decentralized position on the map, probably because the study of sports entrepreneurship as an independent construct is still in its infancy.

Sports entrepreneurship is an emerging field of study, but with great development potential in the field of management. For this reason, some of the studies have focused on the influence that entrepreneurial orientation can have on sports organizations in terms of opportunities to improve performance.

One issue has divided the scientific community with regards to the nature of entrepreneurial orientation [46]: some academics agree that it is a continuous variable, considering its unidimensionality [47,48], while others have stressed its multidimensional nature as a process consisting of several variables [49,50]. Wales et al. [51] analyzed 158 articles on entrepreneurial orientation and found that 98 of them analyzed orientation as a multidimensional construct, consisting of variables such as innovation, proactivity, and risk taking.

Similarly, Lisboa et al. [52] share the thesis of the multidimensional nature of the construct, showing that managers can achieve high levels of performance through various paths that include different combinations and levels of the dimensions that comprise the entrepreneurial orientation.

Escamilla-Fajardo et al. [53] instead consider entrepreneurial orientation both as a unidimensional and a multidimensional construct (consisting of innovation, risk propensity, and proactivity). Their results showed that there is a relationship between all the variables studied, but those that best explain performance in sports clubs are entrepreneurial orientation (understood as a unidimensional construct) and risk propensity (if we consider a multidimensional nature).

The debate is still open, and the substantial differences between the two different approaches are owed to a difference in the weight attributed to the construct. If considered unidimensional, an equal and implicit value is attributed to all the dimensions that make up the Entrepreneurial Orientation; when multidimensional, each variable comes into play in a different way.

In another study conducted in Germany using 22 professional football clubs, Hammerschmidt et al. [10] found that entrepreneurial orientation has a significant and positive impact on both financial and sports performance. Specifically, when the authors analyzed the individual dimensions that make up entrepreneurial orientation, innovation was the only dimension that influenced both types of performance.

These results are in line with previous studies [54,55,56] that show the importance of innovation, regarded as an engine of sports entrepreneurship. 

According to Ratten and Ferreira [57], the concepts of entrepreneurship and innovation are traditionally applied in an entrepreneurial context, with few studies focused on the sports environment; therefore, future research on sports policies should show greater attention to entrepreneurship and innovation, as this would strengthen the field.

The studies presented so far, in addition to highlighting the importance of entrepreneurship (and the constructs associated with it) in the sports field [58,59,60], aim to highlight it as a supporting tool to help make the most of opportunities and resources. They also highlight the need for more studies aimed at clarifying the nature of entrepreneurial orientation, especially in sporting contexts.

#### 3.2.2. Cluster 2: Sport Marketing and Education Role

The second cluster, green, includes 11.3% of the co-occurrence of the keywords: education, entrepreneurship, football, and sport marketing.

In this area, the highest frequency belongs to the terms sport marketing and education. In fact, the most influential articles in this area have shown the importance of entrepreneurial education in sports marketing as an aspect that can provide additional knowledge and, therefore, provide more practical guidance for sports studies [61].

According to Eze [62], there is a growing need to integrate sport with educational and business practices, especially among young people, in order to provide them with all the necessary tools to create new companies in the sports field—a necessary impulse for community growth.

According to Formica [63], entrepreneurship can be represented as a puzzle in which the most important pieces are education, research, application, production, marketing, and sales. Understanding the interlocking of these pieces is also important in the field of sport, in order to compete in a hostile and constantly changing environment.

In fact, with the arrival of new media and the development of technology, sport has become increasingly commercialized, which requires an increase in entrepreneurial marketing initiatives. The companies that have been able to combine these two aspects are the ones that have been most successful in the competitive market [64].

Sports marketing can be defined as the set of sports activities carried out in order to meet the needs of consumers [65]. This is done, specifically, through two different lines of development: “marketing of sport” products, aimed at sports consumers, and “marketing through sport” products, aimed at a wide range of consumers—not specifically sports consumers—such as services used at events with sporting characteristics [65,66,67].

In one study, Afthinos et al. [68] examined whether marketing strategies can promote the diffusion of an Olympic sport nationwide, where results showed great impact of these strategies on its promotion. In particular, small initiatives, such as a graphic design, or a color associated with a club, dramatically increase the interest of followers of a specific team through identification processes and feelings of common belonging.

The importance of sports marketing initiatives has been proven even in small businesses, such as sports clubs. In a study by Gallagher et al. [69], through interviews with managers, the two authors found that marketing practices can help managers to develop and support their clubs. In addition, the development of marketing strategies encourages managers to develop business skills, by helping them expand their networks and grow their clubs thanks to synergy with other small local entities.

Presently, the most visible example of sport as a marketing practice is represented by the diffusion of brands that seem to be important to the sports industry—and specifically for professional football teams, which produce large amounts of money thanks to marketing processes and, consequently, to the support of the fans.

This can be exemplified by Manchester United, which, with more than 100 million followers worldwide [70], is currently considered the most important brand in football, with a value of GBP 259 million [64].

Thanks to continuous globalization, which makes borders very fragile, and technological innovations that eliminate distance, the world is in a constant state of change and transformation; therefore, it seems of vital importance that more academics focus on how to succeed in the combination of sports marketing and international business, by understanding the various social, economic, cultural, and geographical factors that could influence the development of sport worldwide [71].

In this context, entrepreneurial education comes into play as a supporting tool that contextualizes sport in a more realistic dimension, useful in furthering the understanding of the relationships it has with other fields, as well its possible interconnections. In addition, it allows an evolutionary approach to sports studies that takes into account the emerging trends of companies.

#### 3.2.3. Cluster 3: The Relationship between Sport and Social Entrepreneurship: A Tool for Solving Social Problems

Custer 3 has 14.1% of the co-occurrence of keywords, and consists of the following terms: social entrepreneurship, corporate social responsibility, management, and sport for development.

In particular, in this cluster, the emphasis shifts to sport as an agent of social change, especially when the sports organizations are characterized by environments with positive social norms, in which sporting results are secondary to social and democratic values [72].

The relationship between sport and social entrepreneurship can represent a tool for the creation of a society based on democratic and social values.

Finding a universal definition of social entrepreneurship is very difficult due to its complexity and multidisciplinarity [73]. For example, Dacin et al. [74], through a meta-analysis of the literature, found 37 different definitions. A few years earlier, Weerawardena and Mort [75] reached similar conclusions, finding more than 20 different definitions. According to some authors, these different ways of considering social entrepreneurship derive from differences of an economic, cultural, political, and geographical nature [76,77].

In general, the element that distinguishes social entrepreneurship from commercial entrepreneurship is that the latter is driven by economic interests and aims to increase the economic growth of a country, while in social entrepreneurship the focus is on the voluntary sector and non-profit organizations that aim on solving social problems [78,79], which are also recently investing in sports.

Ratten [80], who can be considered a pioneer in this field, defines social entrepreneurship in sport as “the use of social issues to create change in the sporting context. Social entrepreneurship uses sport as a way to encourage solutions to social problems” (p. 561).

In fact, many studies have highlighted the model of sport for development as a strategy used by social entrepreneurs [11], who use sporting activities to, for example, help minority groups, promote social inclusion, aid in socialization processes, and further the resolution of conflicts between cultures. The ultimate goal is to make a social change.

In a study conducted in Spain, Urbano et al. [81] discovered that, although many aspects combine social entrepreneurship with traditional entrepreneurship, social entrepreneurs are driven by a combination of elements and, above all, receive greater support from informal institutions. Specifically, the development of social entrepreneurship has allowed the introduction of new values, such as proactive attitudes to solve problems that affect the community. In this process, the participation of community members becomes essential, both because they are part of the network in which the change will take place, and also because they are the social actors who promote and allow the emergence of these organizations.

In the process of creating social value, understood as the commitment of a company to achieve community improvement [82], corporate social responsibility (CSR) plays an important role as a support element to social entrepreneurship [83], since it helps entrepreneurs to fulfill those responsibilities aimed at improving people’s quality of life [84], and to act positively to strengthen the welfare state [66].

In this sense, many researchers have studied social entrepreneurship and corporate social responsibility as two joint concepts in the field of sports management [11,85,86]. Niño [87] further clarified the nature of the two constructs. According to the author, what unites social entrepreneurship and CSR is the creation of a new social value: both are responses to environmental needs and emerge from a need to provide a decent social system.

In recent years, corporate social responsibility has become an increasingly important priority in the field of sports organizations [88], and many commercial companies are collaborating with sports organizations to offer corporate social responsibility initiatives. One example is Barclays Bank, which in 2007 invested a large sum of money in the Spaces for Sports initiative at Bransbury Park—a program that uses sport to revitalize disadvantaged communities by opening sports facilities.

#### 3.2.4. Cluster 4: Entrepreneurial Intention and Sport

The fourth cluster is the one with the highest number of published articles, representing 46.9% of the co-occurrences. Specifically, the publications that are part of this group contain the following keywords: entrepreneurship, sport, entrepreneurial intention, and student.

In the previous clusters, we have underlined the positive influence that entrepreneurial processes and the constructs associated with them have in the field of sport, in terms of managerial growth and improvement in performance [89,90,91].

In this cluster, the interest of researchers shifted to analyzing variables that can be developed in students, especially in those who have chosen a sport-based curriculum [23,92,93]. This is a cluster that analyzes entrepreneurship in general, and then adapts it to sporting contexts, making it useful for theoretical systematization for the development of the construct.

As Ball [9] points out, the development of entrepreneurial skills is increasingly required in all types of employment, but in the field of sport is essential, owing to its competitive nature. It also facilitates innovation and helps social change [94].

According to Hardy [15], the entrepreneurial spirit provides a new way of viewing sport—a field in constant evolution in today’s society, which gives life to innovative activities.

Ratten [95] pointed out that an athlete’s success is determined by human capital and by individual characteristics. In particular, it has been hypothesized that athletes are characterized by a higher locus of control, situational control, need for achievement, resilience, and discipline than other individuals.

In addition, in a study of Nigerian students, Adeogun et al. [96] found that the development of entrepreneurship in sport is a means to provide employment and a solid tool to fight poverty in the country. The authors recommend that Nigerian institutions change school curricula, with the aim of introducing courses or workshops that can encourage students to engage in an entrepreneurial culture.

However, the awareness that students show of the importance of developing these skills in sports still seems scarce, as they are traditionally linked to a specific type of independent job [97].

For example, a study by Dinning [98], which analyzed the perceptions of 30 students and 5 project hosts on employability and the development of entrepreneurial skills in sport, found that both groups consider project-based learning to be a strategy for improving student skill levels. However, students show a lack of understanding of how to apply these skills in the context of a sports organization.

Despite this lack of perception, especially in students and in local institutions, the importance of entrepreneurial processes in sport as tools for individual and social growth has been widely demonstrated.

However, very little is known about these factors and how they stimulate the promotion of a corporate culture in sport. This is a challenge that international research must accept [17].

#### 3.2.5. Cluster 5: The Effects of Sports Organizations on Social Economic Growth

The fifth cluster is represented by the following keywords: development, social economy, and sports organizations. It constitutes 4.5% of the co-occurrence of the keywords.

The social economy is a term used to describe the set of activities carried out by institutions, organizations, and social actors that regulate their economic activity in various ways through the implementation of projects aimed at social cohesion and solidarity [99]. In fact, the objective of interest of the social economy is the importance attributed to people’s well-being through the implementation of principles of solidarity and social responsibility. In addition, any economic surplus is reinvested in the community to achieve social change or improvement.

According to the “Recent Evolutions of Social Economy” report, carried out by the European Economic and Social Committee in 2017, the European social economy provides more than 13 million paid jobs, equivalent to 6.3% of the active population [100].

In recent years, the sports environment has been reworked as a tool for social and economic growth, especially in developing countries, where social economy organizations have a strong impact as they represent one of the only sources of growth and development [101].

For example, in Africa many associations have emerged that use sport as a tool for raising awareness of HIV among the population.

Although numerous studies [99,102,103] have shown that participation in sport helps bring people from different backgrounds together, and can encourage understanding of others, help change attitudes, and overcome social barriers, nevertheless, uncertainties remain in relation to more lasting change in terms of social inclusion and cohesion, particularly in relation to ethnicity, demonstrating the fact that some barriers require significant and difficult efforts to overcome.

Another limitation of the literature in this cluster concerns the fact that the heterogeneity of sports organizations is not taken into account, i.e., there is a lack of evidence on the differential impacts of different sports on the social economy.

#### 3.2.6. Cluster 6: The Role of Stakeholders in the Development of Sports Enterprises

Cluster 6 is formed by the following keywords: change and stakeholders, and presents 3.4% of the co-occurrence of the keywords. In addition, the word “change” is related to entrepreneurship, while the word “stakeholders” relates to sport.

As can be seen from the literature, change is a constant and characteristic element of the entrepreneurial environment. Globalization, technological innovation, and crisis periods are situations that require companies to adapt in order to cope with changes. For example, with the arrival of new media, sports marketing experts have had to redefine their advertising strategies [104]. The most enterprising companies will be those that can withstand these ambivalent situations through creative skills and, above all, transform them into opportunities and resources for greater entrepreneurial development [105].

Change stimulates the creative and innovative component of a company, transforming it into a dynamic organization. In this sense, a relevant role is assumed for stakeholders—that is, people or groups whose actions directly or indirectly influence the success of sports teams, generating changes [106].

For example, a study by Hoeber and Hoeber [54] on a Canadian sports club found that stakeholders positively influence the innovation capacity of a nonprofit organization, but only if they actively participate in the organization and are interested in creating a significant relationship with it. In particular, stakeholders promote change through knowledge (e.g., analysis of the opportunities and challenges present in a given context, implementation of creative and effective ideas for the solution of problems intrinsic to the organization, and knowledge of how to develop support mechanisms in order to transform ideas into actions).

The strength and relevance assumed by the stakeholders within an organization varies according to the objectives of the company and the challenges it faces [107].

Svensson and Hambrick [108] analyzed how interactions between organizations and stakeholders influence the process of social innovation and, therefore, change. Specifically, the authors conducted a study with 48 leaders on 6 different continents (Africa, Asia, Europe, North America, South America, and Oceania). The results showed that organizations with a strong emphasis on collective learning, a shared disposition to extend the risk of innovation beyond organizational boundaries, and a mutual desire to create new solutions, promote positive social change. In addition, social innovation in these contexts has emerged on three different levels—intra-, inter-, and out-group—as a result of a collective and interactive process between organizations and stakeholders. From the above, the role of stakeholders in the organization seems necessary and useful for the innovation process in sporting organizations, as well as being a means to promote positive social change.

## 4. Discussion

This bibliometric review was carried out to contribute to the systematization of an emerging research field in the scientific landscape—namely, the relationship between entrepreneurship and sport. For this, various bibliometric indicators and cluster analyses were used.

In the research and selection of articles, a validated database was used—Scopus—which produced a total of 239 articles. Some conclusions can be drawn from our analysis.

As the analysis of the bibliometric indicators shows, this is a relatively recent area of study, with its first article dating back to 2004, but one in continuous evolution, reaching its highest number of publications in 2020.

In addition, as was verified by analyzing the scientific journals, this is a research field that covers different areas. Mainly, research on entrepreneurship and sport has been associated with the corporate and managerial research, and the social sciences. However, some journals have also been associated with sports science and economics. This multidisciplinary nature could derive from the fact that sport entrepreneurship as an independent research field is a recent and still evolving topic.

Furthermore, the existing literature on the relationship between entrepreneurship and sport has grown considerably in recent years, especially in the United States, which is the country with the most publications, and in Western, developed societies more generally. Moreover, these data were confirmed by the analysis of the most influential authors in research on sports entrepreneurship. Leading authors were based in Australia (Ratten), Spain (Escamilla-Fajardo, González-Serrano, Moreno, Núñez-Pomar), and the UK (Jones).

These findings suggest that sports entrepreneurship research in developing countries should be encouraged. This would lead to a more balanced picture of sports entrepreneurship practices around the world. This problem is considered important, as management processes are affected by the institutional and cultural contexts in which they are practiced. Therefore, researchers from developing societies could highlight how social, institutional, and cultural contexts influence entrepreneurial processes, which would be useful for consolidating the theoretical framework of this research field.

The results of the bibliometric indicators also highlight two very important aspects that characterize this field of research: only 12% of the authors contributed to more than two articles, and the average is 1.87 authors per article, which underlines the fact that this is a recent and still-evolving field of study, with little collaboration among the authors.

Considering this, an attempt to systematize this area of study seems to be extremely important, and justifies the need for a useful systematic analysis as a source of knowledge. Second, the analysis has shown a lack of quantitative research, as well as theoretical articles, meaning it is important to increase this type of study. This result is in line with the previous reviews [109,110].

The cluster analysis found that most research focuses on a variety of topics. The image is even clearer when the analysis with a minimum of three co-occurrences generated six clusters.

The cluster that obtained the highest percentage of co-occurrences is associated with studies on entrepreneurial intention, especially in students (keywords: entrepreneurship, sport, entrepreneurial intentions, and students). This group accounts for 46.9% of keyword frequency, and is also the area with the most relationships with other groups. This result indicates that at an early stage, researchers were interested in studying which factors stimulate sports entrepreneurship, and how these factors have a positive impact in choosing a sports career.

For these reasons, this cluster has a central position on the map, as it provides theoretical bases for the studies of the other clusters, which instead investigate specific aspects of sports entrepreneurship.

However, there is still no agreement among researchers about the theoretical boundaries of this research field, and most of the studies in this cluster have investigated the same factors that stimulate traditional entrepreneurship. Researchers should take into account the heterogeneity that characterizes sports entrepreneurship, and consider it as an independent analytical construct different from traditional entrepreneurship.

This analysis could be a source of reflection for future research, since using cluster analysis makes it possible to highlight areas in which the research seems to be inadequate.

In fact, our review shows that there are still few studies on the factors that support sports entrepreneurship, such as the roles of education, innovation, and the study of sport from a social perspective. For future research it is important to consider these topics.

Our analysis revealed the great contributions of social entrepreneurship in sports organizations. However, this is an area that is still little explored, towards which many theorizations have been made that have not found solid empirical support. Future research should also accept this challenge.

In addition, as mentioned earlier, it is important to increase quantitative research, which is useful for studying possible constructs related to sports participation, and may help in implementing programs that support young people who wish to pursue sport as a career.

In this sense, many articles have underlined the positive impact of sport entrepreneurship in economic and social growth, in terms of augmenting formal and informal education, changing social behavior and attitudes, promoting the modernization of structures, and increasing understanding of how modern societies function. Sports entrepreneurship is considered an inclusive field, which can develop in various contexts, as it transcends cultural and social settings [96]. However, these data, in addition to lacking empirical support, appear very distant from reality, especially in developing countries, which, as is logical to expect, face challenges that are greater than those faced by developed societies (for example, lack of infrastructure, capital, knowledge, institutional support, etc.). For example, the idea that sport is an easily accessible and useful practice for social mobility is a myth that does not fully correspond to reality [111].

In light of our findings, future research should empirically investigate how the external environment affects the nature and form of sport entrepreneurship. In other words, the literature lacks comparative analyses between countries to further the understanding of how social aspects come into play to mitigate differences and promote inclusion.

It would be appropriate to focus on inequality by race/ethnicity, class, and gender, as well as on the evolving social context of sport, to clarify how sport fits in with other patterns of cultural consumption and the participation of different social groups. More attention should be paid to how leisure products and practices are produced and distributed, and how they intersect with educational, political, and cultural institutions. 

The questions we refer to the “insiders” include: How can sports enterprises contribute to the creation of economic and social value? Are there differences between developed and developing countries in terms of challenges and opportunities? How can social inclusion be achieved through sport entrepreneurship activities? What are the practices of sport entrepreneurship in different societies, and how do they change over time? These are questions that need adequate answers for the implementation of support/development programs for the most disadvantaged groups/contexts.

Lastly, some limitations to this study must be taken into account, specifically from a methodological point of view. First, this work focuses on a group of bibliometric indicators to examine the articles selected for our analysis. Alternative indicators, as well as different databases, like those specialized in entrepreneurship, could be useful in providing a more detailed description of the literature.

In addition, we used cluster analysis, a method widely used in analyses of this type [41,42,43,112], but which provides a limited number of reports that, being based in similarity and co-occurrence techniques, only takes into account the number of keywords considered. This result could represent a limitation considering the high fragmentation that characterizes this research field, as well as its multidisciplinary nature.

## 5. Conclusions

This analysis used the method of scientific mapping of the knowledge base of sport entrepreneurship as a means of delineating the theoretical boundaries of this emerging and, in some ways, still-developing field. It is hoped that the findings of this review will help stimulate and guide researchers approaching this field to provide a basis for the future development of this line of inquiry.

## Figures and Tables

**Figure 1 ijerph-18-04720-f001:**
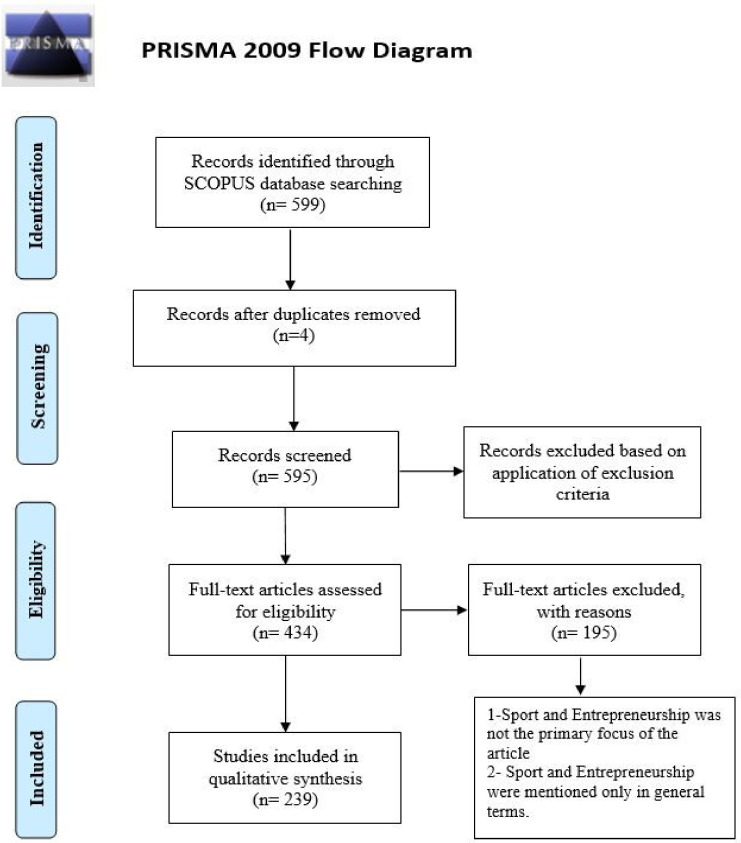
Flow diagram—PRISMA, 2009.

**Figure 2 ijerph-18-04720-f002:**
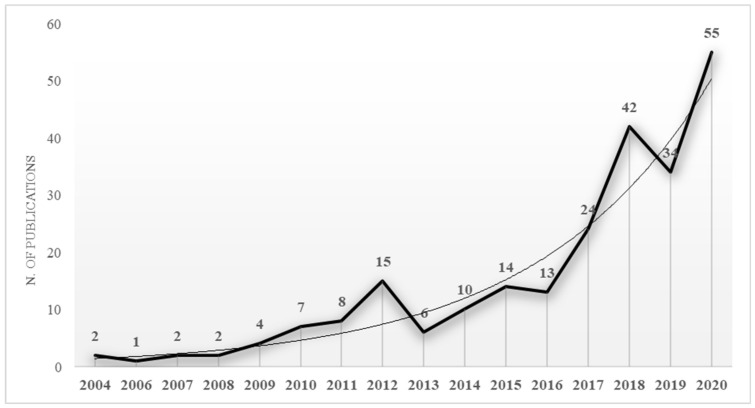
Evaluation of the number of scientific publications per year.

**Figure 3 ijerph-18-04720-f003:**
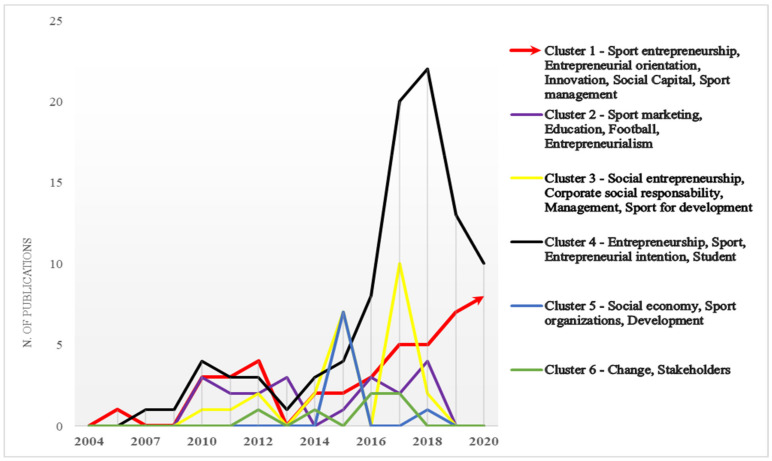
Evaluation of scientific publications per cluster.

**Figure 4 ijerph-18-04720-f004:**
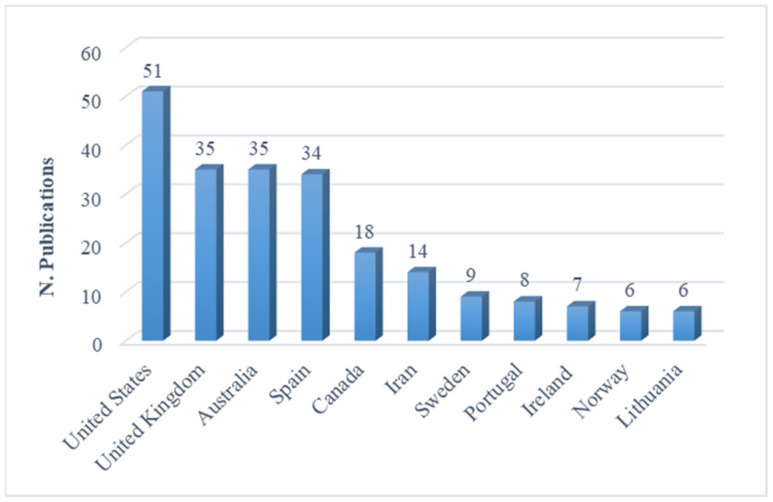
The most productive countries.

**Figure 5 ijerph-18-04720-f005:**
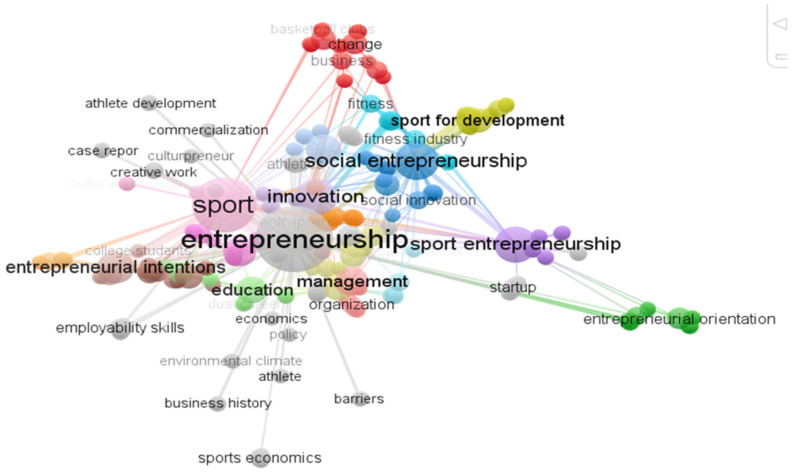
Co-keyword map of the sport entrepreneurship literature (threshold: one co-occurrence).

**Figure 6 ijerph-18-04720-f006:**
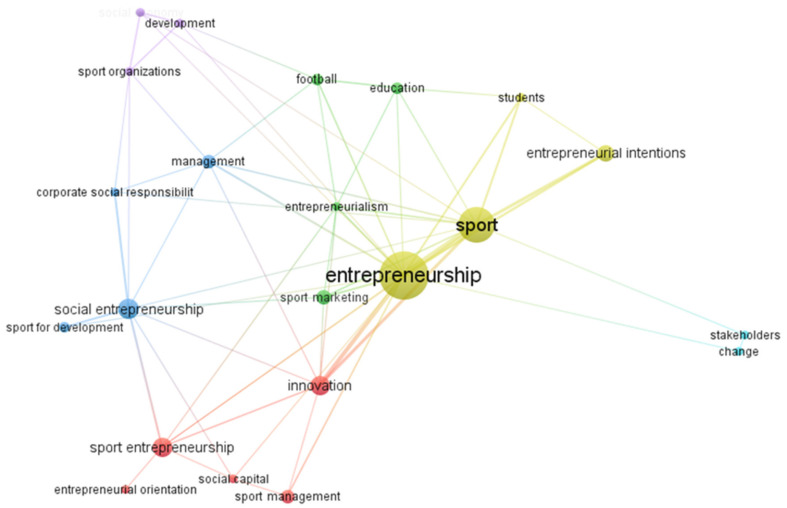
Co-keyword map of the sport entrepreneurship literature (threshold: 3 co-occurrences).

**Table 1 ijerph-18-04720-t001:** Scientific journals with the greatest number of articles published.

R	N. Articles (Out 239)	Journals	Journal Metrics			Research Area
			*h*-Index	JIF	Scopus Citation	
1	16	Sport in Society	34	0.93	368	Sport Sc./Cultural Studies
2	13	Int J Hist Sport	18	0.27	51	Sport Sc./Social Sc.
3	12	Int Entrepreneurship Manag J	50	3.47	207	Business and Manag.
4	10	J Entrepreneurship Public Policy	11	0.60	16	Business and Manag./Social Sc.
5	8	Int J Sport Mang Mark	21	0.55	63	Business and Manag/Sport Sc.
6	7	Sport Management Review	50	3.33	179	Business and Manag/Sport Sc.
7	6	Int J Entrepreneurial Ventur	14	0.43	88	Business and Manag.
7	6	Int J Sport Policy	22	1.98	86	Social Science
8	5	Eur Sport Manag Q	29	1.88	74	Business and Manag/Sport Sc.
8	5	Journal of Sport Management	61	2.35	126	Business and Manag/Sport Sc.
8	5	Retos	6	1.09	10	Social Science
8	5	Sustainability	68	2.59	18	Environmental Sc./Social Sc.

Note. Equally productive journals have the same ranking number.

**Table 2 ijerph-18-04720-t002:** Authors with the greatest number of articles published.

R	N. Articles (Out 239)	Authors	Author Metrics			Affiliations
			h-Index	Scopus Citations	Citation Per Documents	
1	27	Ratten, V.	27	541	20.03	La Trobe Business School, Australia
2	7	Escamilla-Fajardo, P.	3	15	2.14	University of Valencia, Spain
3	6	González-Serrano, M.H.	4	32	5.33	Universidad Católica de Valencia, Spain
3	6	Jones, P.	21	96	16.0	Prifysgol Abertawe, UK
4	5	Moreno, F.C.	13	27	5.4	University of Valencia, Spain
4	5	Núñez-Pomar, J.M.	8	14	2.8	University of Valencia, Spain
4	5	Valantine, I.	6	26	5.2	Lithuanian Sports University, Lithuania

Note. Equally productive authors have the same ranking number.

**Table 3 ijerph-18-04720-t003:** Nature of research and type of sample.

Type of Research	% of the Sample	Type of Sample	% of the Sample
Qualitative Research	44.6%	Sports entrepreneurs/Managers	23%
		Athletes	19.7%
		Students/Youth	9.2%
Quantitative Research	32.6%	Sports Entrepreneurs/Managers	21.9%
		University Sports Students	17.5%
		Athletes	4.6%
Mixed Approach	2.4%	Sports Entrepreneurs	2.3%
		University Sports Students	1.1%
Non-Empirical	28.3%		

**Table 4 ijerph-18-04720-t004:** Summary of the cluster analysis of the “entrepreneurship” and “sport” literature.

Cluster	Keywords	% Articles	Example of Article
1. Sport Entrepreneurship	Entrepreneurial Orientation, Innovation, Sport Entrepreneurship, Sport Management	19.8%	Hammerschmidt et al. (2020). Entrepreneurial Orientation in Sports Entrepreneurship—A Mixed Methods Analysis of Professional Soccer Clubs in the German-Speaking Countries
2. Sport Marketing and Educational Role	Education, Entrepreneurialism, Football, Sport Marketing	11.3%	López-Carril, Villamón, McBride (2020). Social Media in Sport Management Education: Connecting Universities and Sport Industry
3. The Relationship between Sport and Social Entrepreneurship: A Tool for Solving Social Problems	Corporate Social, Management, Social Entrepreneurship, Sport for Development	14.1%	Miragaia, Ferreira, Ratten (2017). Corporate Social Responsibility and Social Entrepreneurship: Drivers of Sports Sponsorship Policy
4. Entrepreneurial Intention and Sport	Entrepreneurship, Entrepreneurial Intention, Sport, Students	46.9%	Lara-Bocanegra et al. (2020). Effects of an Entrepreneurship Sport Workshop on Perceived Feasibility, Perceived Desirability, and Entrepreneurial Intentions: A Pilot Study of Sports Science Students
5. The Effects of Sports Organizations on Social and Economic Growth	Development, Social Economy, Sport Organization	4.5%	Perez-Villalba, Fernandez-Gavira, Caballero-Blanco (2018). The Social Economy in the Sports Entrepreneurship of Spain
6. The Role of Stakeholders in the Development of Sports Enterprises	Change, Stakeholders	3.4%	Pittz T. et al. (2020). Sport Business Models: A Stakeholder Optimization Approach

## Data Availability

Not applicable.

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
