# Peer review of "Entrepreneurship and Sport: A Strategy for Social Inclusion and Change"

_ijerph, 2021, doi:10.3390/ijerph18094720_

Round 1
Reviewer 1 Report
The paper "Entrepreneurship and Sport: a strategy for social inclusion and change" is a very interesting empirical work around entrepreneurship, sports, and relations with social inclusión and change. It adds knowledge to the topic in a relevant area in Leadership, Health and Social Relations.
There are some weaknesses related with the format and language expressions of the paper.
This paper has potential and would be of interest to the audience of International Journal of Environmental Research and Public Health however there are some possible improvements:
- Review minor spell English language in the article, especially in the Abstract.
- Every affirmation should be justified with adequate references.
- The criterion for inclusion of studies is well provided, but some justification of the use of SCOPUS database and not other is needed.
Reviewer 2 Report
The article intents to understand the follow questions: 1) How has its scientific research developed over the years? 2) What are the most productive authors, countries and journals in the scientific landscape? 3) What are the most important issues identified by the existing literature on entrepreneurship in the sports context? 4) What methodological approaches have been used the most? 5) What are the possible gaps currently present in the scientific literature on the relationship between entrepreneurship and sport, and what are its possible lines of interest for future research? Considering this set of questions, the article aims to analyze the scientific production of the theme on screen, under one bibliometric analysis using the scopus database.
With regard to the guiding theme of the article, I perceive it with great complexity. The relationship between entrepreneurship, sport and social inclusion is complex and not peaceful in the literature. The association of sport with social inclusion may even be viable in first world countries, but in the case of underdeveloped countries it is a very distant reality. The sports pyramid is by its exclusive nature, destined for the most skillful, strong, fast, etc. In my view, sport in itself is exclusive, especially when it involves the sports industry and entrepreneurship, the issue gains more dramatic contours, due to the association of sport with the sports industry and the mentality of the capital.
Regarding the merit of the article, I assess that the analysis was very well constructed, with data properly explored and with the guiding questions answered. The data were analyzed in a clear, concise and commented manner. Data interpretation was performed satisfactorily.
As can be seen in the analysis, the article points to a multidisciplinary field, with qualified Journals productions and with significant impact factors. However, it appears that there is a concentration of authors (12% of the authors contributed more than 2 articles) and there is also a concentration of production in certain countries, namely USA, UK, Australia, Spain, and Canada. These data obtained are indicators of the exclusion process to which I referred previously. The highlights and the concentration of production in countries like USA, UK and Australia are quite revealing and should be better addressed by the authors of the article.
Need to make a correction on the line
351: Cluster 3 has 14.1% …
I have no doubt that, to paraphrase the authors:
“The relationship between sport and social entrepreneurship can represent a tool for the creation of a society based on democratic and social values”. However, it is not what actually occurs. Perhaps even among first world countries this can be achieved, but in the case of underdeveloped countries, this reality is far from being "real".
“In addition, in a study with Nigerian students, Adeogun et al. [93] found that the development of entrepreneurship in sport is a means to provide employment and a solid tool to fight poverty in the country. The authors recommend that Nigerian institutions change school curricula with the aim of introducing courses or workshops that can encourage students to engage on an entrepreneurial culture”. At this point it would be interesting to ask yourself how many Nigerian children / adolescents have access to school or have been socially included through sport? So, in this case, in fact the lack of quantitative and qualitative studies, equally pointed out by the authors of the article, makes sense.
I think that these aspects discussed above could be added to the conclusion.
I recommend the following work as a reference:
COAKLEY, Jay. Sports in Society: issues and controversies. MacGraw Hill.
Reviewer 3 Report
The article is interesting because it will help to stimulate and guide researchers to approach and develop this field of research. This analysis has been made possible by the method of scientific mapping of the knowledge base on sport entrepreneurship. However, minor changes should be made to the manuscript before publication:
The authors should revise the numbering of citations, because some are missing in the text: 33, 40, 52, 56, 63,...
Conclusion section
It is a bit short, so it could be expanded a bit to be able to make concrete conclusions.
References section
The numbering of the references needs to be revised, because there are many citations that do not appear in the text.
Reviewer 4 Report
- Rewrite abstract with subheadings that explain the "A brief statement of method" and "results with significant values" Please modified the Abstract.
- The conclusion should be the core section of a paper. However, it seems too simple in this paper.
- This paper many be has obtained research results through literature collation. Unfortunately, your paper was found not suitable for the International Journal of Environmental Research and Public Health.
- The manuscript offers very little new insight. I’ve learned very little from reading this manuscript. Thus, I don’t see major contribution from the manuscript which needs further development and draws more focuses. The structure of the paper should be reorganized as well. Due to the above concerns, I would like to suggest ‘resubmit’ this paper to other journal.
Author Response
Thanks for your valuable comment and suggestions to enhance the overall quality our the manuscript. In the revision, we have tried our best to address all the issues raised by you (in the revised manuscript they are underlined in red), both regarding the abstract and the conclusions.
Best Regards
Round 2
Reviewer 4 Report
- Identified gap in current work then a research design, method and or objectives that help fill in the research gap.
- The conclusion should be the core section of a paper. However, it seems too simple in this paper.
- This paper many be has obtained research results through literature collation. Unfortunately, your paper was found not suitable for the International Journal of Environmental Research and Public Health.
- The author did not modify the manuscript of according to the reviewer comments. In addition, IJERPH is an SSCI grade Journal. The manuscript offers very little new insight. I’ve learned very little from reading this manuscript. Thus, I don’t see major contribution from the manuscript which needs further development and draws more focuses. The structure of the paper should be reorganized as well. Due to the above concerns, I would like to suggest ‘reject’ this paper.
